# Identification and Characterization of Common Bean (*Phaseolus vulgaris*) Non-Nodulating Mutants Altered in Rhizobial Infection

**DOI:** 10.3390/plants12061310

**Published:** 2023-03-14

**Authors:** Rocío Reyero-Saavedra, Sara Isabel Fuentes, Alfonso Leija, Gladys Jiménez-Nopala, Pablo Peláez, Mario Ramírez, Lourdes Girard, Timothy G. Porch, Georgina Hernández

**Affiliations:** 1Centro de Ciencias Genómicas, Universidad Nacional Autónoma de México, Avenida Universidad 2001, Cuernavaca 62210, Morelos, Mexico; maroresa@ccg.unam.mx (R.R.-S.); sasa@ccg.unam.mx (S.I.F.); leija@ccg.unam.mx (A.L.); gjimenez@ccg.unam.mx (G.J.-N.); ppelaez@ccg.unam.mx (P.P.); mario@ccg.unam.mx (M.R.); girard@ccg.unam.mx (L.G.); 2USDA-ARS, Tropical Agriculture Research Station, 2200 P.A. Campos Avenue, Suite 201, Mayaguez 00680, Puerto Rico; timothy.porch@usda.gov

**Keywords:** nodulation, *Phaseolus vulgaris*, EMS, infection, rhizobia, symbiosis

## Abstract

The symbiotic N_2_-fixation process in the legume–rhizobia interaction is relevant for sustainable agriculture. The characterization of symbiotic mutants, mainly in model legumes, has been instrumental for the discovery of symbiotic genes, but similar studies in crop legumes are scant. To isolate and characterize common bean (*Phaseolus vulgaris*) symbiotic mutants, an ethyl methanesulphonate-induced mutant population from the BAT 93 genotype was analyzed. Our initial screening of *Rhizobium etli* CE3-inoculated mutant plants revealed different alterations in nodulation. We proceeded with the characterization of three non-nodulating (*nnod*), apparently monogenic/recessive mutants: *nnod*(1895), *nnod*(2353) and *nnod*(2114). Their reduced growth in a symbiotic condition was restored when the nitrate was added. A similar *nnod* phenotype was observed upon inoculation with other efficient rhizobia species. A microscopic analysis revealed a different impairment for each mutant in an early symbiotic step. *nnod*(1895) formed decreased root hair curling but had increased non-effective root hair deformation and no rhizobia infection. *nnod*(2353) produced normal root hair curling and rhizobia entrapment to form infection chambers, but the development of the latter was blocked. *nnod*(2114) formed infection threads that did not elongate and thus did not reach the root cortex level; it occasionally formed non-infected pseudo-nodules. The current research is aimed at mapping the responsible mutated gene for a better understanding of SNF in this critical food crop.

## 1. Introduction

A key to the success of the legume family, comprising around 700 genera and accounting for one-third of the primary crop production in the world, was the evolution of symbioses with mycorrhizal fungi and nitrogen-fixing bacteria, collectively known as rhizobia, that facilitate plant nutrient acquisition [1,2,3]. The symbiotic N_2_-fixation (SNF) process in the legume-rhizobia interactions is ecologically and economically relevant for sustainable agriculture; it is a valuable source of nitrogen, which is an essential element for crop production [4]. This symbiotic process, occurring in N-limited soils, takes place in root-developed specialized organs called nodules. Nodulation and SNF are complex processes that are tightly regulated, both in rhizobia and in its specific legume host. The first step in this interaction is the communication between the two symbionts through molecular signals, namely flavonoids secreted by the plant root to the rhizosphere that trigger the compatible rhizobia biosynthesis of lipochitooligosaccharides, known as nodulation factors (NFs). NFs’ perception by the plant, through its binding to high-affinity receptors mainly in the tip of emerging root hairs (RH), results in a series of molecular responses leading to rhizobial infection and nodule organogenesis. The infection responses start with the attachment of rhizobia to the legume root, mainly to the RH tip, giving rise to several pre-infection responses such as RH deformations, swelling and curling, bacterial entrapment in the so-called infection chamber and the formation of an infected thread (IT)—an invasive invagination of the plant cell—and the invasion of the cortical cells. Rhizobia infection and nodule organogenesis are two connected processes, each one controlled by different genes in time and space [2,5,6]. Proceeding the rhizobia infection, the root cortex cell division is activated to develop the nodule primordia. Moreover, intricate signaling pathways are activated, involving fine regulation of nodule genes to develop mature nodules with functionally differentiated bacteroids [2,5,7,8].

Advances in legume genomics and forward or reverse genetic research, especially during the last 20 years, have contributed to the understanding of legume genes required for the effective symbiosis with rhizobia [6,9]. For such valuable knowledge, genetic approaches on legume mutants resulting in the identification and mapping of their mutated genes responsible for alterations in different stages of the rhizobia symbiosis have been essential. Thus, mutant populations, generated with chemical or physical mutagens, have been obtained from different legume crops, such as the pea [10,11,12], soybean [13,14,15], alfalfa [16] and common bean [17,18,19,20,21], as well as from the model legumes *Lotus japonicus* [22,23,24] and *Medicago truncatula* [25,26]. More recently, mutant collections of model legumes, resulting from transposon insertions, have been obtained [27,28,29,30] and are publicly available. These resources have been instrumental for increasing our understanding about plant genes required for SNF, providing a catalog of nearly 200 genes that also includes information from the crop species soybean (*Glycine max*) and common bean [6]. 

Common bean, a diploid species with 11 chromosomes, is the most important grain legume for direct human consumption worldwide. It is the principal source of non-animal protein from over three hundred million people, mainly throughout Latin America and Africa [4]. However, common bean genomic/genetic research has been left behind because of problems such as the lack of an effective transformation protocol to generate stable transgenic mutant plants with transposon insertions. Since the 1990s, common bean mutant populations, induced by the chemical mutagen ethyl methanesulphonate (EMS), were obtained and screened for alterations in the nodulation process [17,19,20]. Two non-nodulating common bean mutants have been analyzed regarding some genetic and physiological characteristics [31,32,33], but their mutated genes have not been mapped nor identified. To our knowledge, there are only two *P. vulgaris* nitrate-tolerant supernodulating mutants: R32 (derived from the OAC RICO variety) and SV45 (derived from the Swan Valley variety) [19], where the mutated gene responsible for the alteration in the Autoregulation of Nodulation (AON) process has been mapped, identified and molecularly characterized [34]. 

It is evident that the isolation, mapping and genetic/physiological/molecular characterization of common bean symbiotic mutants are lacking. With the aim to isolate new common bean rhizobia-symbiotic mutants, a mutant population of 1692 M_4_ lines, developed by Porch et al. [21], was screened in this study. This mutant population, generated for the targeted and induced local lesion in the genome (TILLING), was derived from the *P. vulgaris* reference genotype BAT 93 and developed at the Center for Tropical Agriculture (CIAT), and was the first genome sequenced (549.6 Mb) from the Mesoamerican genetic pool [35]. The screening of the same mutant population [21] allowed the isolation of several putative *lpa* (low phytic acid) mutants; one of these was characterized as an additional lpa1 allele [36,37]. 

The aim of this work was to screen the EMS-generated BAT 93 mutant population [21] to identify and characterize mutants with defects in the rhizobia SNF process. We hypothesized that it would be possible to obtain mutants with impairment in a different stage of the symbiosis. Our objective for this work was to phenotypically characterize the selected symbiotic mutants, with stable altered phenotypes, by analyzing the presence or absence of known structures developed in different steps of the symbiosis, such as rhizobial infection organogenesis. 

Our screening of the mutant lines inoculated with *Rhizobium etli* CE3 revealed lines with different alterations in nodulation. We characterized three selected mutant lines, appearing to be monogenic and recessive, showing a stable non-nodulating phenotype. A phenotypic analysis of these mutants evidenced distinct alterations in each one of them that affected the different early steps of the rhizobial infection process: the rhizobia-induced root hair deformation, the infection chamber formation and infection thread development.

## 2. Results

### 2.1. Screening for Altered Nodulation Phenotype in the Mutagenized Common Bean Population

In this work, 1692 M_4_ lines from an EMS mutant population of the wild type (WT) BAT 93 *P. vulgaris* genotype [21] were screened for rhizobia-symbiosis mutants. Seeds from each line were inoculated with *R. etli* CE3 [38]. After 4 weeks, when the nodulation is expected to be at the maximum, mutant plants were carefully removed from their pots and their nodulation capacity was recorded after a visual comparison with that of BAT 93 wild type (WT) plants. In the initial screening, 93 lines with an altered nodulation phenotype, rated as an increased nodulation, decreased nodulation or no nodulation, were selected. In a second screening performed to confirm their nodulation-impaired phenotype, most of the mutant lines showed a wild-type nodulation phenotype. Thus, we decided to further analyze 36 mutant lines that showed a non-nodulation phenotype, by exploring the segregation of nodulation (wt) vs. non-nodulation (*nnod*) in the M_5_ progeny. We screened 9 to 12 M_5_ plants derived from each of two M_4_ plants of the selected mutant lines (Appendix A). The majority─23 out of 36 lines analyzed─showed normal nodulation (wt). The remaining 13 lines showed variable wt:*nnod* segregation, ranging from 2:1 to 7:1. The mutant lines, 08IS-1895, 08IS-2353 and 08IS-2114, showed the closest 3:1 segregation expected for a monogenic and recessive mutation. Therefore, we selected these three mutant lines, hereafter denominated *nnod*(1855), *nnod*(2353) and *nnod*(2114), for further phenotypic analysis. 

### 2.2. Growth, Root Phenotype and Nodulation Capacity of Nnod Mutants

Contrasting with the WT plants, the *R. etli* CE3-inoculated *nnod*(1895), *nnod*(2353) and *nnod*(2114) mutant plants showed symptoms of nitrogen deficiency, such as reduced growth and chlorotic leaves (Figure 1A,B). However, the growth of the three mutant lines was fully restored when a sufficient amount of nitrate was supplied (Figure 1B). The latter indicates that the growth impairment of inoculated *nnod* mutants was due to the lack of symbiotic nitrogen fixation.

Seedlings from the WT and from each of the *nnod* mutants were inoculated with three different rhizobia strains, which are effective and common bean symbionts: *R. etli* CE3 [38], *Sinorhizobium fredii* NGR234 [39] and *Rhizobium phaseoli* CIAT652 [40]. Each bear a construct for the constitutive *GUS* gene expression. After 25 days post-inoculation (dpi), the root growth and nodulation capacity were evaluated. 

The WT, *nnod*(1895) and *nnod*(2353)-inoculated plants demonstrated similar root growth, while *nnod*(2114) demonstrated about a 70% decrease in primary root length, regardless of the rhizobia strain used (Figure 2A,B and Appendix A).

As expected, the WT plants demonstrated effective nodulation with each of the rhizobia inoculum (Figure 3A). The *nnod*(1895) and *nnod*(2353) mutant plants were unable to develop nodules when inoculated with any of the rhizobia strains used (Figure 3A), thus indicating that their *nnod* phenotype resulted from the mutation and was not dependent on the rhizobia inoculum. The *nnod*(2114) plants did not form nodules when *S. fredii* NGR234 or *R. phaseoli* CIAT652 were used as inoculum. However, in *R. etli* CE3-inoculated *nnod*(2114) plants, a few white nodules, or pseudo-nodules, could occasionally be observed (Figure 2A and Figure 3A). These pseudo-nodules were 50% smaller than nodules formed in the BAT 93 plants (Figure 3B). The GUS reporter gene signal revealed a full *R. etli* CE3 infection in WT nodules that was evident even in small and immature nodules (Figure 3C,D). Contrastingly, no *R. etli* CE3/pGUS infection was detected in *nnod*(2114) pseudo-nodules (Figure 3C,D).

Our results demonstrated that from the three common-bean *nnod* mutants analyzed, only *nnod*(2114) mutant plants inoculated with *R. etli* CE3 were capable of forming empty, non-functional pseudo-nodules.

### 2.3. Comparative Microscopical Analysis of Inoculated BAT 93 and Nnod Mutants’ Inoculated Roots at the Early Stages of the Symbiotic Process

In order to gain information about the symbiotic process alterations of the analyzed *nnod* mutants, a detailed microscopic evaluation of the morphological features that distinguish the different steps of rhizobia infection, during the early symbiotic stage, was carried out. The morphological features of the primary host infection were examined by light microscopy after the GUS-staining of *R. etli* CE3/pGUS 25 dpi roots.

The rhizobial infection process starts with the rhizobia attachment to the root, mainly RH tips, that triggers pre-infection responses such as RH swelling and curling, which are required for the rhizobia colonization [2]. The evaluation of the RH deformation revealed a similar amount of characteristic and effective deformed or curled RH in WT, and *nnod*(2353) and *nnod*(2114)-inoculated roots (Figure 4A). However, the *nnod*(1895)-inoculated roots demonstrated decreased RH curling and a higher number of non-effective or “spatula-like” [41] RH deformations (Figure 4A,B). These results (Figure 4) indicate that the *nnod*(1895) mutant was impaired in the initial step (RH curling) of the symbiotic pathway, which blocks signal transduction pathways triggered by rhizobia.

In the next step of rhizobia infection, bacteria are enclosed in the curled RH pocket known as the infection chamber or microcolony, which allows for the formation and development of IT [42]. A difference in the infection chamber formation was observed in the *nnod*(2353)-inoculated roots. Though the *nnod*(2353)-inoculated roots formed a similar number of effective RH curling as WT and *nnod*(2114) (Figure 4A), its GUS-stained roots demonstrated a high amount of small, blue dots that higher microscope magnification identified as infection chambers (average of 7.3 infection chambers per root cm) that were arrested at this step and thus unable to develop into normal IT (Figure 5). The absence of visible blue dots in WT or *nnod*(2114)-inoculated roots indicates that in these plants, the normally formed infection chambers were not arrested and developed to the next steps of the symbiotic pathway: the formation of the IT.

As mentioned above, the *nnod*(2114) was the only common bean mutant whose plants form empty pseudo-nodules after *R. etli* CE3 inoculation (Figure 3), thus indicating that these mutant plants did recognize rhizobial signaling, were infected by rhizobia and partially activate nodule organogenesis. Our root hair deformation analysis demonstrated a slight increase in the effective root hair formation in *nnod*(2114) compared to WT-inoculated plants (Figure 4A). The analysis of formed IT in the WT demonstrated its normal development, with the IT reaching the root cortex cell layer, while the *nnod*(2114)-inoculated roots only showed the truncated IT unable to reach the root cortex (Figure 6A). In addition, the analysis of 10 dpi roots revealed an increased number of truncated IT formed in the *nnod*(2114) compared to normal IT formed in WT roots (Figure 6B). These results indicate that *nnod*(2114) plants were able to recognize rhizobia, as evidenced by the effective RH deformation and infection chamber formation, but the development of the fully formed IT was impaired, and the IT were thus incapable of reaching the root cortex; a few of these abnormal infections may progress into empty pseudo-nodule formation.

## 3. Discussion

The symbiosis between plants and nitrogen-fixing bacteria is a unique model of an intimate interaction that has major ecological impacts. The advent of plant genomics and refined genetic resources has contributed to the current understanding of various essential protagonists of this complex symbiosis. Particularly in the model legumes, *M. truncatula* and *L. japonicus*, more than 200 symbiotic genes have been identified [6]. However, for agriculturally important legume crops such as *P. vulgaris*, such relevant genetic information remains scant. In this work, we analyzed an EMS mutant population of the BAT 93 genotype, developed by Porch et al. [21], to identify and characterize mutants affected in the rhizobia N2-fixing symbiosis. Using this approach, non-nodulating common bean mutants were identified several years ago [17,20,31,33]; however, the precise physiological characterization identifying a specific impaired symbiotic or genetic analysis identifying the responsible mutated genes have not been reported. In addition, Park and Buttery [19] identified the nitrate-tolerant common bean super-nodulating mutants; these have been recently molecularly and genetically characterized by Ferguson et al. [34]. Based on previous research on the soybean, using comparative—and functional-genomics approaches—two alleles of the common bean gene *PvNARK* (NODULE AUTOREGULATION RECEPTOR KINASE) were identified as responsible for the supernodulation phenotype of two common bean mutants in each of the R32 and SV45 cultivars [34]. The soybean and common bean NARK genes, as well as orthologous genes from different legumes, code for specialized LRR-receptor kinases that are key components of the AON, a systemic signaling pathway that controls the nodule number. The AON begins with the production of root-specific CLE peptides that are transported via the xylem to the shoot and are perceived by NARK which, in turn, leads to the production of a shoot-derived inhibitor signal that is transported to the root, where it acts to inhibit further nodulation events [5,7,43]. To our knowledge, this is the only fully characterized common bean rhizobial-symbiotic mutant.

Our initial screening of the *R. etli* CE3-inoculated M_4_ common bean mutant plants demonstrated several lines with altered nodulation, such as increased, decreased or no nodulation, as compared to the WT BAT 93-inoculated plants. However, most of these mutants nodulated normally in the subsequent generations, thus indicating that a large number of plants were erroneously selected on the basis of nonheritable characteristics. Thus, we proceeded to characterize three mutant lines with a non-nodulating phenotype (*nnod*), indicative of a monogenic, recessive mutation in each line and that has been stable up to the M_9_ generation. The diminished growth rate observed in the inoculated *nnod*(1855), *nnod*(2353) and *nnod*(2114) mutant plants, as compared to the WT BAT 93 plants, was restored when plants were grown in a non-symbiotic condition, with optimal nitrate content added. This result, which is in agreement with results from the common bean non-nodulating mutants previously analyzed [17,20,44], led us to conclude that the *nnod*(1855), *nnod*(2353) and *nnod*(2114) mutants can assimilate and utilize inorganic N for normal plant growth, but their inability to nodulate prevents the N fixation and growth under rhizobia symbiosis conditions. 

Plants from *nnod*(1855) and *nnod*(2353) mutant lines displayed the non-nodulating phenotype across the three different rhizobia species (*R. etli* CE3, *S. fredii* NGR234 and *R. phaseoli* CIAT652) used as inoculum; these are effective symbionts for WT common bean plants [38,39,40]. Occasionally, a few small/white pseudo-nodules were observed in *nnod*(2114) mutant plants inoculated with *R. etli* CE3 but not with the other rhizobia species. The observed *nnod*(2114) pseudo-nodules were empty without rhizobia invasion. These results, also reported for the previously analyzed common bean mutants [31], indicate that the non-nodulating phenotype results from the mutation in each mutant line and not from the rhizobia inoculum used. 

In this work, we went beyond the observation of the presence/absence of nodules in the non-nodulating mutants to discover at which stage of the rhizobia infection or nodule organogenesis processes were impaired. Our phenotypic analysis revealed variations among the mutants, indicating that each mutant bears a different genetic defect, blocking different steps of the rhizobial infection process, as depicted in Figure 7.

The current knowledge about legume genes relevant for the rhizobia symbiosis [6] provides a comprehensive set of genes that were classified according to their participation in different symbiotic stages, such as early signaling, host range restriction and rhizobial infection. This information, as well as the phenotypes of legume mutants impaired in early symbiotic genes and their similarity with the phenotype of the common bean symbiotic mutants here reported, allowed us to propose possible candidates of mutated genes causal for the defects observed in each mutant. 

Contrasting with RH deformations observed in the WT and the other *nnod* mutants, the *nnod*(1895) mutant demonstrated less effective RH curling, which results from the altered rhizobia communication and initial interaction with the legume root [2], and a higher amount of “spatula-like” non-effective RH deformations. The RH showing aberrant—ineffective—deformations are unable to trap rhizobia and to form the infection chamber required for allowing the rhizobia entry via the IT formation and development [41,45]. Our results indicate that the common bean *nnod*(1895) mutant is capable of the rhizobia recognition and is impaired in the very early steps of rhizobia infection (Figure 7). The analysis of model legume mutants revealed that defects in specific NF-receptors do not allow compatible rhizobia recognition by the legume root; thus, no RH deformation is observed [46]. Instead, if legume genes participating in the rhizobia infection signaling pathway are mutated, a phenotype of the decreased effective RH curling deformation and increased ineffective RH deformation has been observed. The possible candidate genes that we propose for the *nnod*(1895) mutant are: SYMRK (SYMBIOSIS RECEPTOR-LIKE KINASE) [47], NUP133, NUP85 (NUCLEOPORINS) [48,49], ERN (ETHYLENE RESPONSE FACTOR REQUIRED FOR NODULATION) [45] and AGO5 (ARGONAUTE 5) [41]. The mutants *Ljsymrk, Ljnup85, Ljnup133* and *Mtern1/ern2* double mutant are non-nodulating mutants that show increased and non-effective—“spatula like”, swelling and branching—RH deformation, and no infection chambers or IT development [41,45,47,48,49]. In addition, *P. vulgaris* roots silenced for AGO5 showed an increase in the “spatula-like” RH deformation [41].

The *nnod*(2353) and *nnod*(2114) mutants are impaired in the rhizobial infection process (Figure 7). These mutants are able to form RH effective deformations, which are important for trapping rhizobia, allowing bacteria to divide and form a microcolony known as the infection chamber [6,42]. In addition, the *nnod*(2114) forms IT, but only truncated ITs are observed in the epidermis layer, being unable to develop and reach the root cortex cell layer. Few empty pseudo-nodules were rarely formed in the *R. etli* CE3-inoculated *nnod*(2114) roots, but not when *S. fredii* NGR234 or *R. phaseoli* CIAT652, the compatible symbionts, were used as the inoculum. A similar phenotype was reported for the *Ljnup133* mutant that formed few ineffective white nodules when inoculated with two compatible rhizobia, but not with a third strain tested [48]. We propose the following as possible candidate genes’ causal for the rhizobial infection defects of the *nnod*(2353) and *nnod*(2114) mutants: EPR (EXOPOLYSACHARIDE RECEPTOR) [50], CYCLOPS [51], CBS (CYSTATHIOINE-β-SYNTHASE-like domain-containing) [52], FLOT4 (FLOTILIN) and REM1 (REMORIN) [53], NPL (NODULATION PECTATE LYASE GENE) [54], the ARPC1 (ACTIN-RELATED PROTEIN COMPONENT 1) [55], SCARN (SUPPRESOR OF cAMP RECEPTOR DEFECT—NODULATION) [56], NAP (NCK-ASSOCIATED PROTEIN1) and PIR (121F-SPECIFIC P53 INDUCIBLE RNA) [57] genes that participate in the actin remodeling required during rhizobial infection, VPY (VAPYRIN) [58], LIN(LUMPY INFECTIONS)/CERBERUS [59], RPG (RHIZOBIUM-DIRECTED POLAR GROWTH) [60] and IEF (INFECTION-RELATED EPIDERMAL FACTOR) [61] genes that are necessary for the IT polar growth into the cortex. The model legume mutants impaired in one of these rhizobial infection genes show a similar phenotype-effective RH deformation, infection chamber formation and development of ITs that are truncated, branched and not progressed beyond the epidermis—as those are observed for common bean *nnod*(2353) and *nnod*(2114) mutants. Some of the model legume mutants are also capable of starting nodule organogenesis, forming small, white, empty pseudonodules [50,54,57,58,59,60,61], similar to the *nnod*(2114) mutant. A hypothesis to explain this is that the signal (NFs) sent by rhizobia in the IT is necessary and sufficient to trigger this process [51].

Contrasting with WT and the other two *nnod* mutants, a short root phenotype was observed in the *nnod*(2114) mutant plants. This phenotype could be associated with the *nnod* phenotype, as in the *LjNap1* and *LjPir1* mutants with a shorter root phenotype that was observed but not further analyzed [57]. Another possibility is that the short root phenotype is caused by another unrelated mutated gene present in the *nnod*(2114) mutant line, something that is common in mutant populations generated by EMS treatment. The analysis of short root and *nnod* phenotypes in individuals from the progeny of a back cross of *nnod*(2114) × BAT 93 will distinguish between these two possibilities, revealing whether the two phenotypes co-segregate or not. 

The current research in our group focuses on identifying the mutated gene of *nnod*(1985), *nnod*(2353) and *nnod*(2114), which is responsible for the rhizobial symbiosis impairment in each mutant. For this, we consider the strategies of the whole genome sequence or targeted exon capture/sequencing. The whole genome sequence approach, using appropriate algorithms for a bioinformatic analysis, has been successfully used to identify a mutated gene of interest in different plants. For, example Addo-Quaye et al. [62] reported the identification of a mutated gene participating in GA biosynthesis using this approach in EMS-generated *Sorghum bicolor* mutants. We propose to do a comparative genome sequence analysis between genome sequences from gDNA pools from wt or *nnod* individuals obtained from the backcross BAT 93 × *nnod* mutant. The targeted exon capture/sequencing [63] has proven to be successful for different plant species. The whole exome sequence implies a considerable initial investment to build the required platform. However, we plan to use this method, focusing in exomes of the candidate mutated genes that we have proposed for our *nnod* mutants.

The achievement of the identification of the causal mutated genes will contribute to a better understanding of the relevant genes for the rhizobia infection process in the common bean. In addition, symbiotic mutants have been instrumental for identifying genes that participate in different related regulatory pathways of the rhizobia symbiosis [6]. An example of the latter has recently been published [64] by using the only common bean symbiotic mutant: *Pvnark* [34]. The analysis of *Pvnark* and *Gmnark* mutants, using physiological, reciprocal grafting and split-root approaches, determined that the AON pathway is required for the inhibition of the nodulation by phosphorus deficiency [64]. Thus, the identification of the mutated genes in the common bean non-nodulating mutants analyzed in this work would allow for the use of these mutants as a tool to answer research questions about rhizobial symbiosis regulation in the most important legume for human consumption [4].

## 4. Materials and Methods

### 4.1. Plant Material and Growth Conditions

The common bean (*P. vulgaris*) BAT 93 wild type genotype, representative of the Mesoamerican gene pool [35], was used in this work. This genotype was developed at the International Center for Tropical Agriculture (CIAT, Cali Colombia) and derived from a double cross involving four Mesoamerican genotypes.

The screening of symbiotic mutants was performed in 1692 M_4_ lines of the EMS mutant population developed by Porch et al. [21] in the BAT 93 background. These seeds were bulk-harvested from plants grown in the field in Puerto Rico. 

Seeds were surface sterilized in 10% V/V commercial sodium hypochlorite for 5 min and rinsed with distilled water. Then seeds were germinated on sterile moistened filter paper at 30 °C in darkness for two days. Germinated seedlings were planted in pots with wet sterile vermiculite/agrolite (1:1) and grown in glasshouse with semi-controlled conditions (temperature 25–30° C, 70% humidity and natural illumination). 

For SNF evaluation immediately after planting, plantlets were inoculated with 1 mL of saturated liquid culture of the desired rhizobial strain (see below), applied directly to the root. Plants were watered, once a week, with 100 mL per plant N-free (50%) Summerfield nutrient solution [65] and subsequently with tap water 2 times per a week. For the non-symbiotic condition, a full nutrient Summerfield solution (7 mM N-content) was used to water the plants. 

### 4.2. Screening for Symbiotic Mutants

Two seeds from each M_4_ mutant line were used for the initial screening. Two seeds were surface sterilized, germinated and planted in 20 cm diameter pots with sterile vermiculite, with two seeds per line in each pot. In addition, pots with two WT BAT 93 seeds each, were included for a comparison of nodulation capacity. Plants were inoculated with *Rhizobium etli* CE3 strain, as described. After 28 days of growth, the plants were uprooted and visually inspected for nodulation, comparing mutated lines with WT BAT 93 plants. Plants of mutated lines with visible changes in their symbiotic phenotype were re-planted in the same pots and grown to maturity. To boost seed production, NPK-granulated fertilizer was added and the full nutrient Summerfield solution was used for watering. The M_5_ seeds were collected from each line and two seeds per line were re-screened to confirm the stability of their altered nodulation phenotype. 

### 4.3. Rhizobial Strains and Culture Conditions

*R. etli* CE3/pGUS, *S. fredii* NGR234/pGUS and *R. phaseoli* CIAT652/pGUS strains were obtained by triparental mating using the *Escherichia coli* DH5α strain-carrying plasmid pGUS (pFAJ1700 derivative expressing the *uidA* gene (gus) under the lacZ promoter, Tc^R^) as a donor [66] and the *E. coli* DH5α strain-carrying plasmid pRK2013 as a conjugation helper, Km^R^ [67]. Transconjugants carrying plasmid pGUS were selected as Fm and Tc resistant derivates. 

*Rhizobium* strains were grown at 30 °C in a peptone–yeast (PY) medium supplemented with CaCl_2_ (7 mM) and the appropriate antibiotic [68]. *E. coli* strains were grown at 37 °C in the Luria-Bertani (LB) medium supplemented with the appropriate antibiotic. Antibiotics were used at the following concentrations: fosfomycin (Fm), 100 µg mL^−1^ (*Rhizobium*); tetracycline (Tc), 10 µg mL^−1^ (*E. coli* and *Rhizobium*); and kanamycin (Km), 30 µg mL^−1^ (*E*. *coli*). A modified Eckhardt procedure was used to determine the plasmid profiles in the *Rhizobium* derivative strains [69].

### 4.4. Observation (Analyses) of Infection and Nodulation Events

WT and *nnod* mutants were analyzed after 25 dpi to examine the symbiosis phenotype. Five plants per genotype were inoculated with different strains of rhizobia. In order to probe whether the rhizobia colonized and established a symbiosis with the plants, roots and nodules were immersed in a GUS staining solution (0.05% 5-bromo-4-chloro-3-indolyl-β-d-glucuronic acid, 100 mm sodium phosphate buffer (pH 7), 0.5 mm potassium ferrocyanide, 0.5 mm potassium ferricyanide, 10 mm Na2EDTA and 0.1% Triton X-100) and incubated for 3 h at 37 °C. Roots were then cleared in diluted 10% NaClO for 10 min, and rinsed in phosphate buffer. Nodule images were taken using a ZEISS SterREO Discovery microscope and measured with a digital Vernier caliper. 

### 4.5. Root Analysis

The main root at 25 dpi in WT and *nnod* mutant plants was measured from digital pictures. These images were processed with SmartRoot software [70]. Fifteen plants of each phenotype were analyzed. 

The root hair phenotype was analyzed with a bright field microscope (ZEISS, Axioskop 2). Two segments of the root from WT and *nnod* mutants were taken after GUS staining for counting the number of effective or non-effective root hairs per cm. Two plants per genotype were analyzed. 

### 4.6. Statistical Analysis

All results are reported were analyzed by one-way ANOVA followed using Dunnett’s multiple comparisons test (*p* < 0.05) or one-tailed *t* tests (*p* < 0.05) with the software GraphPad Prism version 9.4.1 for Windows, GraphPad Software, San Diego, CA, USA (www.graphpad.com) (accessed on 1 September 2022). 

## 5. Conclusions

In the present study, the EMS-generated mutant population from the BAT 93 *P. vulgaris* genotype [21] was screened for mutants altered in the rhizobia SNF process. Several mutant lines, inoculated with *Rhizobium etli* CE3, revealed different symbiotic alterations. From these, three mutant lines with a stable non-nodulating (*nnod*) phenotype, indicative of a single recessive mutation, were characterized. Their phenotypic analysis revealed an impairment in a different step of the rhizobial infection process in each of the mutants. The *nnod*(1895) mutant is impaired in root hair deformations, which is an initial step after the plant sensing of the Nod-Factors rhizobial signal. The *nnod*(2353) mutant showed normal root hair curling and rhizobia entrapment to form infection chambers but was blocked at this step. The *nnod*(2114) formed infection threads that did not elongate and thus did not reach the root cortex level; it occasionally formed non-infected pseudo-nodules. Several possible candidate genes for each of the mutants were proposed. 

## Figures and Tables

**Figure 1 plants-12-01310-f001:**
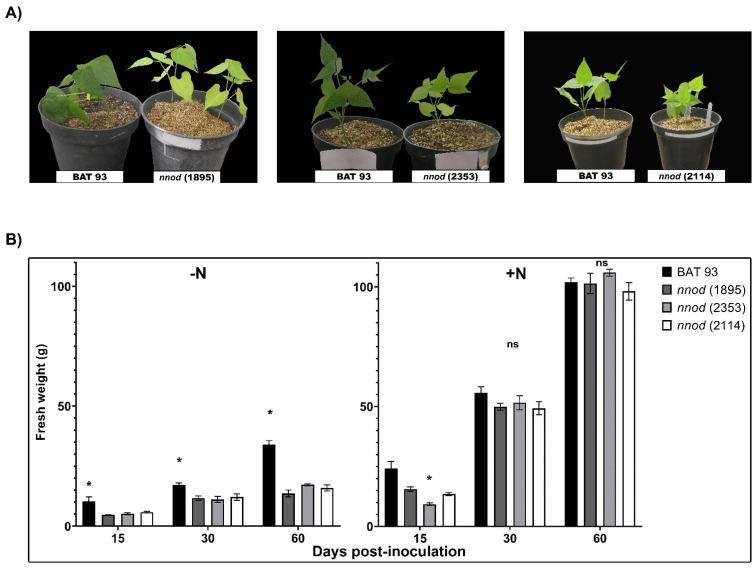
Phenotype of the *Phaseolus vulgaris* non-nodulating mutants: *nnod*(1895), *nnod*(2353) and *nnod*(2114). (**A**) Representative plants of wild type (WT) BAT 93 (left) and each of the non-nodulating (*nnod*) mutants (right), after 25 days post-inoculation (dpi) with *Rhizobium etli* CE3. (**B**) Growth, calculated as fresh weight (g), of WT in comparison to *nnod* mutants, watered with N-free solution and inoculated with *R. etli* CE3 (left) or watered with full-nutrient solution (7 mM N-content, right). The data are presented as average from at least 5 individual plants with standard error (SE) as indicated by error bars. (*) significant difference in fresh weight between WT and each *nnod* mutant-inoculated root; (ns) no significant difference (*p* ≤ 0.05, Student’s *t*-test).

**Figure 2 plants-12-01310-f002:**
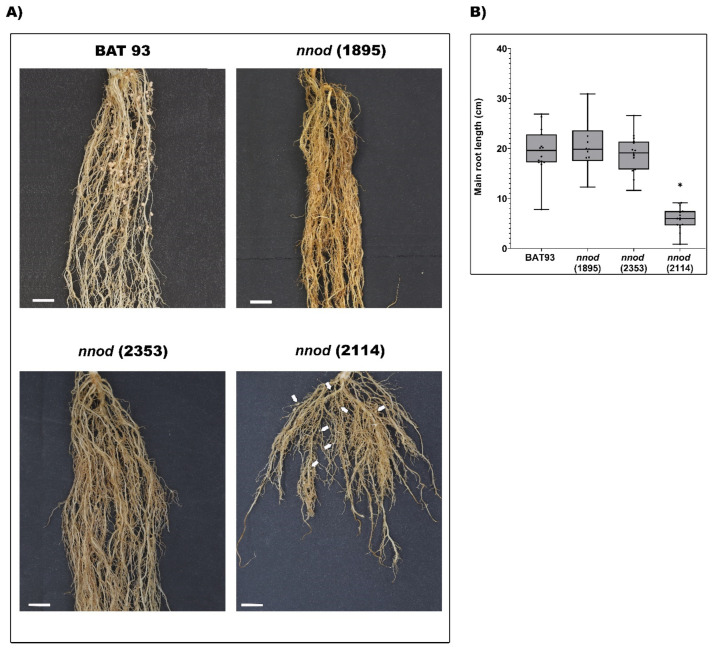
Nodulation and root growth of *P. vulgaris* BAT 93 and *nnod* mutants. (**A**) BAT 93 and *nnod* mutants after 25 dpi with *R. etli* CE3. White arrows point to the small-white pseudo-nodules formed in *nnod*(2114)-inoculated plants. Bar = 1 cm. (**B**) Length of the main root of WT and *nnod* mutant plants from at least 5 inoculated individuals per genotype. In box plots, horizontal box side represents the first and third quartile while the outside whiskers represents the minimum and maximal values. (*) significant difference of main root length between WT and each *nnod* mutant-inoculated root (*p* ≤ 0.0001, Student’s *t*-test).

**Figure 3 plants-12-01310-f003:**
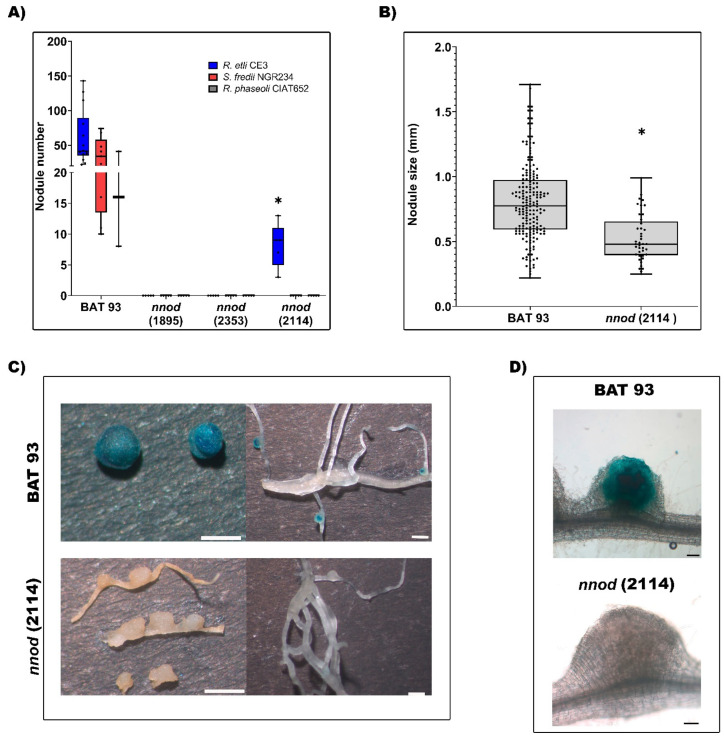
Nodulation phenotype of WT and *nnod* mutants. Analyses were conducted at 25 dpi. (**A**) Quantification of nodules formed in WT or *nnod* mutants inoculated with *R. etli* CE3, *Sinorhizobium fredii* NGR234 or *Rhizobium phaseoli* CIAT652 strains. (**B**) Evaluation of the perimeter of nodules formed in WT and *nnod*(2114) plants inoculated with *R. etli* CE3 in each genotype. Nodule number and size data (**A**,**B**) were obtained from at least 10 individual plants. In box plots, horizontal box side represents the first and third quartile, while the outside whiskers represent the minimum and maximal values. (**C**,**D**) Colonization of *R. etli* CE3/pGUS in nodules of WT or *nnod*(2114), visualized after GUS staining. C. Bar = 1 mm. D. Bar = 100 μm. (*) significant difference (*p* ≤ 0.1, Student’s *t*-test).

**Figure 4 plants-12-01310-f004:**
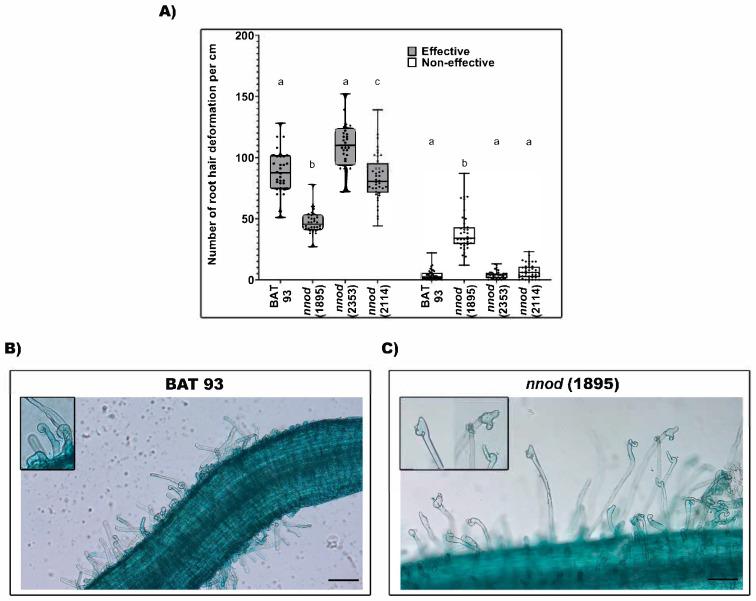
Response to rhizobia infection of WT and *nnod*(1895) mutant inoculated with *R. etli* CE3/pGUS. (**A**) Rhizobia-induced, effective or non-effective, root hair deformations were quantified at 25 dpi. Two biological replicates per genotype were analyzed by counting the root hair deformation in 20 root segments (1 cm) from each inoculated root. In box plots, horizontal box side represents the first and third quartile, while the outside whiskers represent the minimum and maximal values. Different lower-case letter from each set of values (effective and non-effective root hair deformations) indicates significant difference according to one-way analysis of variance (ANOVA) (*p* ≤ 0.0001). Representative image of rhizobia-induced root hair deformations in WT-inoculated roots; insert shows root hair curling and hook-type effective root hair deformations (**B**) and in *nnod*(1895)-inoculated roots; insert shows aberrant spatula-like non-effective root hair deformations (**C**). Bar = 100 µm.

**Figure 5 plants-12-01310-f005:**
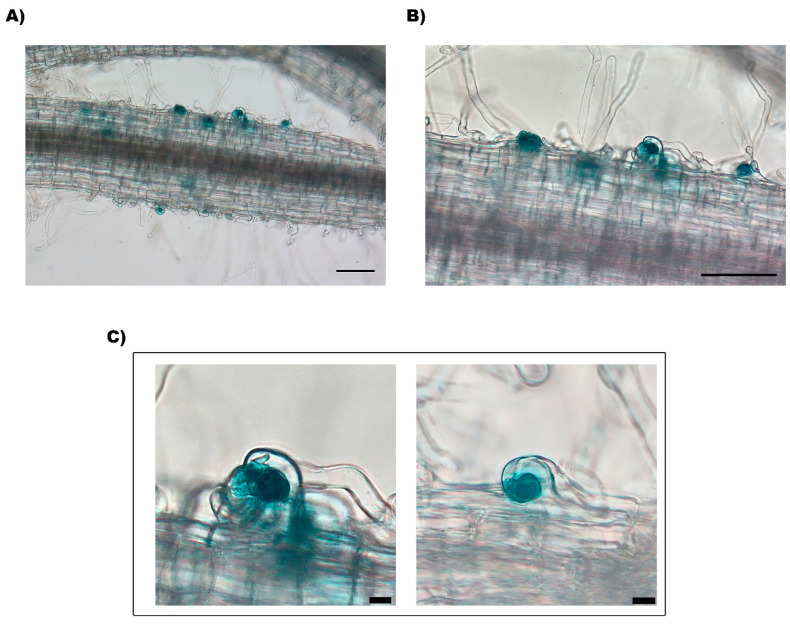
Infection phenotype of *nnod*(2353) mutant inoculated with *R. etli* CE3/pGUS: formation of infection chambers. Microscopic observations of the infection chamber, or microcolony, formed in curled root hair of inoculated *nnod*(2353) roots, visualized after GUS staining. (**A**,**B**) Bar = 100 μm. (**C**) Bar = 10 μm.

**Figure 6 plants-12-01310-f006:**
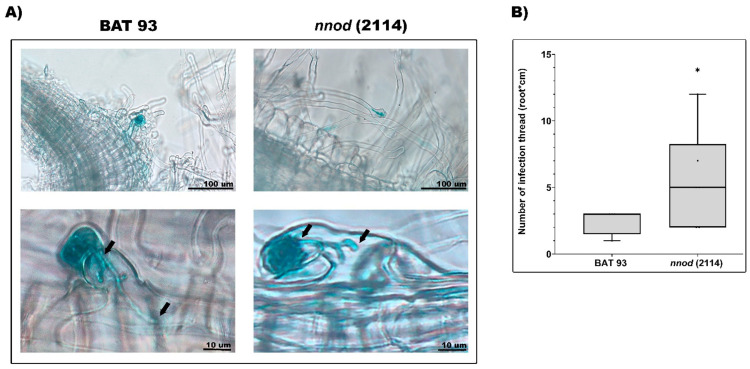
Infection phenotype of *nnod*(2114) mutant inoculated with *R. etli* CE3/pGUS: formation and development of infection threads (ITs). (**A**) Microscopic observations of the ITs formed in BAT 93 and in the *nnod*(2114) mutant, visualized after GUS staining. Arrows point to progression of an IT in the WT genotype (left) and abortion of IT development in the *nnod*(2114) mutant (right). (**B**) Quantification of ITs in 5 root segments (1 cm) of WT and *nnod*(2114)-inoculated roots. In box plots, horizontal boxes represent the first and third quartiles, while the outside whiskers represent the minimum and maximal values. (*) Significant difference of IT number between WT and *nnod*(2114)-inoculated roots (*p* ≤ 0.1, Student’s *t*-test).

**Figure 7 plants-12-01310-f007:**
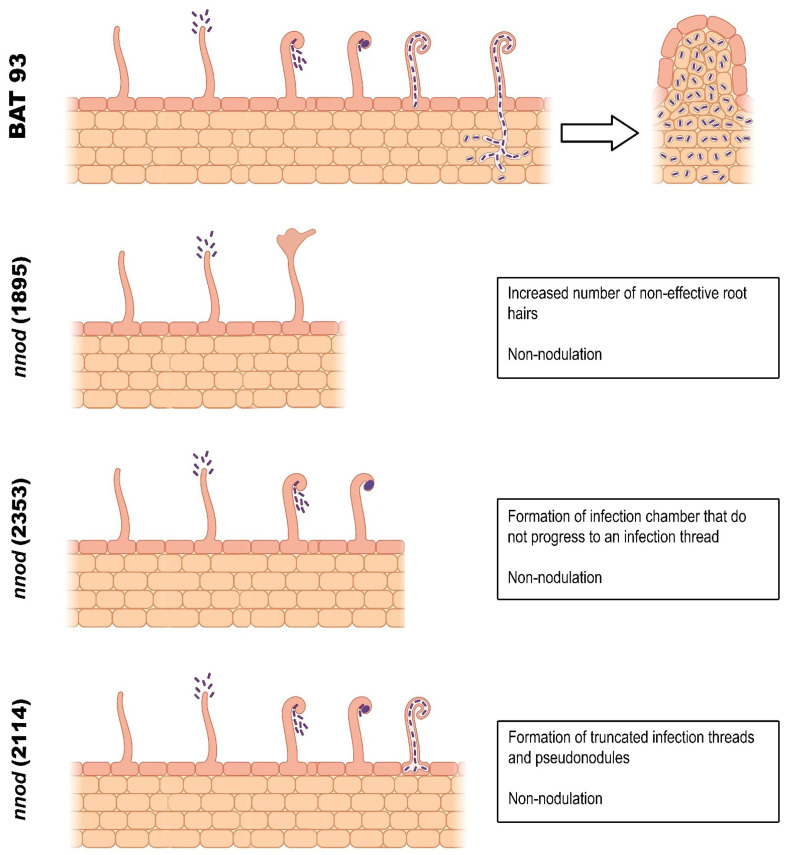
Summary of the defective phenotype of *P. vulgaris* non-nodulating mutants: *nnod*(1985), *nnod*(2353) and *nnod*(2114), impaired in a different step of the rhizobia infection process. This figure was created with Biorender.com.

## Data Availability

Data is contained within the article or Appendix A.

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
