# Peer review of "Identification and Characterization of Common Bean (Phaseolus vulgaris) Non-Nodulating Mutants Altered in Rhizobial Infection"

_plants, 2023, doi:10.3390/plants12061310_

Round 1
Reviewer 1 Report
This manuscript describes the characterization of three interesting non-nodulating mutants of common bean, altered in the first steps of the interaction. Authors followed the correct steps to properly identify the altered steps in the symbiotic process and shows adequate microscopic evidences of the affected phenomena. I think this manuscript merit publication after minor but necessary changes.
First of all, I could not see figures 2A, 3A, 3B and 4A at all. The quality of this images is really low and it was impossible to see what authors represent with this figures. Also Figure 1B, that in addition lacks of necessary statistics. Are differences in dry weight significant?
There are sentences in results that could be moved to discussion section o expressed in past tense, as result are usually written. Some changes suggested:
-Lines 119-120: The non-nodulation phenotype in rhizobia-inoculated plants from selected mutant lines was stable from M5 to M9 generations.
-Lines 126-127: (Fig 1B.), suggesting that growth impairment of inoculated nnod mutants was due to…
-Line 192: indicated… was impaired…
-Lines 233-237: this paragraph should be moved to discussion section or expressed in past tense if it is maintained in results section: indicated… were able to recognize rhizobia…
-Line 251: use Particularly instead of Mainly
Finally, the discussion section could finish with a short explanation of the authors suggesting the strategy that they will follow to identify the genes mutated in the three mutants described in this work, since they consider this fact of interest at the end of the discussion.
Author Response
We acknowledge your comments and suggestions that, indeed, helped to improve the manuscript.2
Point 1. First of all, I could not see figures 2A, 3A, 3B and 4A at all. The quality of this images is really low and it was impossible to see what authors represent with this figures. Also Figure 1B, that in addition lacks of necessary statistics. Are differences in dry weight significant?
REPONSE 1.
We apologize for the low quality of the figures appearing in the manuscript’s template you received for review. Before v.1 submission, we had prepared good quality figures, according to the requirements from “PLANTS”. However, we now realized that when we pasted each Figure in the journal template (.doc) the figures, indeed, diminished their quality considerably and we interpret this was the version that you and other Reviewers received. Although, upon submission, we separately uploaded the good quality figures, we interpret that these were not sent to you. Anyway, for the revised version, we have further improved the quality of the figures and we have modified some figure panels (graphs) trying to make them clearer, this are: Fig.1B, Fig. 2B, Fig.3A, B, Fig. 4A, and Fig. 6B. As suggested, in Figure 1B we have included data on significance difference among the weight values.
In addition, I have commented, the problem of the figures’ quality after pasting these in the template, with our Managing Editor and he offered to revise and be sure that the Figures in the manuscript’s template that will be send for revision conserve the quality from the original Figures we are re-submitting or he will send you separate figures with good quality. Thus, we hope this problem will be solved.
POINT 2.
We modified the sentences you mentioned as suggested, the specific modification for each one is:
There are sentences in results that could be moved to discussion section o expressed in past tense, as result are usually written. Some changes suggested:
-Lines 119-120: The non-nodulation phenotype in rhizobia-inoculated plants from selected mutant lines was stable from M5 to M9 generations
RESPONSE 2A.
The sentence was deleted from “Results”, since we realized that a very similar one appears in “Discussion”.
-Lines 126-127: (Fig 1B.), suggesting that growth impairment of inoculated nnod mutants was due to…
-Line 192: indicated… was impaired…
-Lines 233-237: this paragraph should be moved to discussion section or expressed in past tense if it is maintained in results section: indicated… were able to recognize rhizobia…
RESPONSE 2B, 2C, 2D:
The sentences were left in “Results” and the verbs were modified to past tense
Line 251: use Particularly instead of Mainly
RESPONSE 2E:
- The sentence was modified as suggested.
POINT 3. Finally, the discussion section could finish with a short explanation of the authors suggesting the strategy that they will follow to identify the genes mutated in the three mutants described in this work, since they consider this fact of interest at the end of the discussion.
RESPONSE 3.
In the Discussion section, we added information about the strategies we propose to follow to identify the causal mutated symbiotic gene in each mutant.
Reviewer 2 Report
Dear authors,
The manuscript entitled "Identification and Characterization of Common bean (Phaseolus vulgaris) non-nodulating mutants altered in rhizobial infection." presents an interesting research on one of the most important symbiotic process for plants. The research is presented explicitly and all the aspects are discussed.
There are some general comments and suggestions that I consider will improve the quality of your work:
Make a separate paragraph at the end of the introduction with the aim and your hypotheses/objectives. Each in a separate sentence.
Add a conclusion section, where to point the most important findings.
Improve the quality of your figures, they are very interesting and need to be clear.
I have enjoyed the reading and the form of your work, and I consider it important for future researches.
Author Response
We acknowledge your comments and suggestions that, indeed, helped to improve the manuscript.
POINT 1.
Make a separate paragraph at the end of the introduction with the aim and your hypotheses/objectives. Each in a separate sentence.
RESPONSE 1. We have added a paragraph in the introduction stating the aim, hypothesis and objectives of this work.
POINT 2.
Add a conclusion section, where to point the most important findings.
RESPONSE 2. The conclusion section has been added.
POINT 3.
Improve the quality of your figures, they are very interesting and need to be clear.
RESPONSE 3.
We apologize for the low quality of the figures appearing in the manuscript’s template you received for review. Before v.1 submission, we had prepared good quality figures, according to the requirements from “PLANTS”. However, we now realized that when we pasted each Figure in the journal template (.doc) the figures, indeed, diminished their quality considerably and we interpret this was the version that you and other Reviewers received. Although, upon submission, we separately uploaded the good quality figures, we interpret that these were not sent to you. Anyway, for the revised version, we have further improved the quality of the figures and we have modified some figure panels (graphs) trying to make them clearer, this are: Fig.1B, Fig. 2B, Fig.3A, B, Fig. 4A, and Fig. 6B. As suggested, in Figure 1B we have included data on significance difference among the weight values.
In addition, I have commented, the problem of the figures’ quality after pasting these in the template, with our Managing Editor and he offered to revise and be sure that the Figures in the manuscript’s template that will be send for revision conserve the quality from the original Figures we are re-submitting or he will send you separate figures with good quality. Thus, we hope this problem will be solved.
Reviewer 3 Report
The authors describe the isolation and preliminary characterization of non-nodulation mutants of Phaseolus vulgaris, common bean, an important crop legume for human consumption. The experimental work was straightforward and clearly presented, though a bit short of results. To identify three non-nodulating mutants of bean is nice but the interesting thing is which genes are they defective in. In the current version of the manuscript, I would discuss more in detail what are the possible candifdate genes for the three mutants. The review by Roy and co-authors (Plant Cell, 2020) provides a comprehensive list of genes required for symbiotic nodule development and functioning to select from. Similarly, I miss the complete discussion of the shorter root phenotype of nnod 2114. In addition, If the mutations are in known symbiotic genes, the results on the genetic defects will be rather difficult to publish. So, in the place of the authors, I would make an RNA sequencing or targeted RNA/genomic sequencing on the mutants to identify the mutations and to strengthen the current manuscript.
Author Response
We acknowledge your comments and suggestions that, indeed, helped to improve the manuscript, specially to enrich the Discussion.
POINT 1. In the current version of the manuscript, I would discuss more in detail what are the possible candifdate genes for the three mutants. The review by Roy and co-authors (Plant Cell, 2020) provides a comprehensive list of genes required for symbiotic nodule development and functioning to select from.
RESPONSE 1.
In the Discussion section we have added details about our proposed candidate causal mutated genes for each mutant. For this we considered the similarities of our P. vulgaris mutant phenotypes with phenotypes of reported for symbiotic mutants impaired in genes involved in the rhizobia infection process. Such information was obtained from the review paper of Roy et al. 2020 and other papers there referred.
POINT 2. Similarly, I miss the complete discussion of the shorter root phenotype of nnod 2114.
RESPONSE 2. We added discussion about the root phenotype observed for the nnod(2114) mutant.
POINT 3. In addition, If the mutations are in known symbiotic genes, the results on the genetic defects will be rather difficult to publish. So, in the place of the authors, I would make an RNA sequencing or targeted RNA/genomic sequencing on the mutants to identify the mutations and to strengthen the current manuscript.
RESPONSE 3. Thanks very much for your proposal “taking our (the authors’) place”. My group and Tim Porch’s group would like very much to publish, asap, this paper because although the mutant population was obtained long ago there is only one publication about its analysis (Cominelli et al, 2018; lpa mutant). As suggested by another reviewer, in the Discussion section we added information on the 2 approaches we propose to follow for the identification of the causal mutated gene in each mutant. In fact, fyi, the whole genome sequencing approach is quite advanced; collaborating with bioinformatic experts from my Center we are now analyzing the whole genome-seq results focusing in high impact variants, that may tell us the information we want. We did not include an RNA-seq approach because, as you might know, the main gDNA change induced by EMS mutagenesis is a GC to AT base pair change. Even though this could be high impact change, I.e. generating stop codons, this changes will not be easily identified from RNA-seq data as opposed, for example, from mutagen-genera. ted deletion in a mutant population. We will begin a targeted-exon sequencing approach, based in the mutated genes we proposed soon - we cannot afford a whole exome analysis- depending in the results from whole genome sequencing. Certainly, you may be right about the difficulty to publish a mutant in an already known symbiotic gene, however this would depend in the phenotype we will find in such common bean mutant. As we all now, there may be gene neo-functionality resulting from the evolution of each legume species and we may find this scenario. In addition, we can use that mutant – in a known symbiotic gene- to answer as different question about its function, similar as what we comment in the Discussion about using the NARK P. vulgaris mutant to answer an unknown function. This resulted in original knowledge about PvNARK role in nodule control of nodule number in a phosphate deficiency stress condition (Isidra-Arellano et al., 2018). Anyway, after -hopefully- publishing this first paper on characterization of P. vulgaris symbiotic mutants, we would deal with the possible problem to publish about their identified causal gene... when time comes
Round 2
Reviewer 3 Report
The manuscript was clearly improved by adding more detailed discussion regarding the known genes potentially affected in the mutants. As different mutant alleles of a single gene or even variations in a homogenous population might result in diverse severity of the phenotype, the authors might combine the discussion on the potentially affected genes in nnod(2353) and nnod(2114) because both phenotypes can be observed in the mutants mentioned mentioned in the manuscript and also in the rpg (Arrighi, 2008) or ief (Kovacs, 2022) mutants.
Author Response
POINT 1: ....the authors might combine the discussion on the potentially affected genes in nnod(2353) and nnod(2114) because both phenotypes can be observed in the mutants mentioned.
RESPONSE 1. We have combined the proposed candidate muatted genes for both mutants: nnod(2353) and nnod(2114).
POINT 2. ....and also in the rpg (Arrighi, 2008) or ief (Kovacs, 2022) mutants.
RESPONSE 2. These candidate genes were added as weil as the correspondoing references.